# Extracellular Vesicles Derived from Human Umbilical Cord Mesenchymal Stromal Cells as an Efficient Nanocarrier to Deliver siRNA or Drug to Pancreatic Cancer Cells

**DOI:** 10.3390/cancers15112901

**Published:** 2023-05-24

**Authors:** Florian Draguet, Nathan Dubois, Cyril Bouland, Karlien Pieters, Dominique Bron, Nathalie Meuleman, Basile Stamatopoulos, Laurence Lagneaux

**Affiliations:** 1Laboratory of Clinical Cell Therapy (LCCT), Jules Bordet Institute, Université Libre de Bruxelles (ULB), 90 Rue Meylemeersch, 1070 Brussels, Belgium; 2Department of Haematology, Jules Bordet Institute, Université Libre de Bruxelles (ULB), 90 Rue Meylemeersch, 1070 Brussels, Belgium; 3Medicine Faculty, Université Libre de Bruxelles (ULB), Route de Lennik 808, 1070 Brussels, Belgium

**Keywords:** mesenchymal stromal cells, umbilical cord, extracellular vesicles, nanocarrier, siRNA, drug, cancer

## Abstract

**Simple Summary:**

Pancreatic ductal adenocarcinoma (PDAC) is one of the most lethal cancers worldwide. Treatment of PDAC remains a major challenge. Recently, exosomes derived from mouse skin fibroblasts were modified to deliver siRNA to specifically target mutant KRAS. This approach resulted in disease suppression and increased overall survival in an animal model. This study aims to evaluate, in vitro, the use of human umbilical cord mesenchymal stromal cell (UC-MSC)-derived extracellular vesicles (EVs) to deliver molecules (siRNAs or drugs) to specifically target pancreatic cancer cells.

**Abstract:**

Pancreatic ductal adenocarcinoma (PDAC) is one of the most lethal cancers worldwide. Treatment of PDAC remains a major challenge. This study aims to evaluate, in vitro, the use of human umbilical cord mesenchymal stromal cell (UC-MSC)-derived EVs to specifically target pancreatic cancer cells. EVs were isolated from the FBS-free supernatants of the cultured UC-MSCs by ultracentrifugation and characterized by several methods. EVs were loaded with scramble or KRAS^G12D^-targeting siRNA by electroporation. The effects of control and loaded EVs on different cell types were evaluated by assessing cell proliferation, viability, apoptosis and migration. Later, the ability of EVs to function as a drug delivery system for doxorubicin (DOXO), a chemotherapeutic drug, was also evaluated. Loaded EVs exhibited different kinetic rates of uptake by three cell lines, namely, BxPC-3 cells (pancreatic cancer cell line expressing KRAS^wt^), LS180 cells (colorectal cell line expressing KRAS^G12D^) and PANC-1 cells (pancreatic cell line expressing KRAS^G12D^). A significant decrease in the relative expression of the KRAS^G12D^ gene after incubation with KRAS siRNA EVs was observed by real-time PCR. KRAS^G12D^ siRNA EVs significantly reduced the proliferation, viability and migration of the KRAS^G12D^ cell lines compared to scramble siRNA EVs. An endogenous EV production method was applied to obtain DOXO-loaded EVs. Briefly, UC-MSCs were treated with DOXO. After 24 h, UC-MSCs released DOXO-loaded EVs. DOXO-loaded EVs were rapidly taken up by PANC-1 cells and induced apoptotic cell death more efficiently than free DOXO. In conclusion, the use of UC-MSC-derived EVs as a drug delivery system for siRNAs or drugs could be a promising approach for the targeted treatment of PDAC.

## 1. Introduction

The incidence of pancreatic adenocarcinoma (PDAC) is increasing. PDAC is expected to become the second leading cause of cancer-related death by 2030. PDAC has an extremely poor prognosis [1], and PDAC treatment remains a major challenge. PDAC is refractory to most treatments. The desmoplastic stroma is an important biological barrier to drug delivery and activity. Several relevant mutations have been identified. The main mutation that occurs in PDAC, accounting for approximately 90% of mutations in PDAC, is the KRAS mutation. The principal substitution mutation in KRAS is G12D [2]. Oncogenic substitution at residue G12 leads to the constitutive activation of KRAS. This mutation results in the stimulation of downstream signaling partners (PI3K-AKT-mTOR or RAF-MEK-ERK), which in turn cause increased proliferation, decreased apoptosis, altered metabolism and modifications of the tumor microenvironment [3]. These features make KRAS one of the most attractive targets in cancer biology. Numerous attempts have been made to inhibit KRAS and its downstream signaling pathways in PDAC. However, no therapeutic agents that directly target KRAS have been clinically approved [4]. Specifically targeting, via siRNA, mutant KRAS could be a promising approach for PDAC treatment [5]. Unfortunately, siRNA therapy is limited by short serum half-lives, inadequate delivery to specific sites, intracellular digestion and transient therapeutic targets [6]. Different compositions of nanoparticles (NPs) have been tested to target KRAS and its associated pathway via siRNA. However, none of these NPs have translated to clinical application [7]. EVs are endogenous carriers for delivering active molecules (proteins, RNA) that can transfer information targeting cancerous cells or genes implicated in cancer development. In PDAC, EVs are implicated in a multitude of pathways including the induction of chemoresistance [8]. While EVs have been largely described as tumorigenesis promoters, it is conceivable that EVs also possess antitumor functions and act to restrain disease progression. Various studies demonstrated that EVs play a significant role in the treatment of PDAC by reducing its progression and aggressiveness. For example, EVs can transfer curcumin in pancreatic cancer cells to increase the in vitro cytotoxicity [9]. The use of EVs to deliver drugs to pancreatic cancer cell lines allowed the restoration of gemcitabine sensitivity [10]. The inhibition of some genes in association with exosomal microRNAs can inhibit cancer progression in PDAC [11]. As reported by Xiong J. et al. [12], EVs may be excellent tools to prevent the chemoresistance of PDAC. An innovative strategy for the treatment of pancreatic cancer that uses engineered exosomes to target the KRAS^G12D^ gene has been proposed [13]. Kamerkar et al. modified exosomes derived from mouse skin fibroblasts to deliver siRNA to specifically target mutant KRAS, resulting in disease suppression and increased overall survival in mouse models. An ideal cell source of EVs for the delivery of siRNAs or drugs would be one that produces an abundance of nonimmunogenic EVs. Recently, human-MSC-derived EVs have been proposed as drug delivery vehicles for chemical drugs, genetic materials or proteins [14]. MSCs are multipotent cells that reside in various tissues. MSCs have been investigated as therapeutic alternatives not only due to their regenerative potential but also due to their immunomodulatory effects and anti-inflammatory properties. Moreover, MSCs from different tissues exert differential effects on tumor progression, which suggests that the parent source of EVs is an important factor that must be investigated in the development of therapeutic strategies [15]. We chose umbilical cords as a source of MSCs due to several advantages compared to other sources, namely, that umbilical cords are medical waste and are widely available and that umbilical-cord-derived MSCs are easy to collect, exhibit rapid self-renewal, exhibit low immunogenicity and lack tumorigenicity.

This study aims to specifically target pancreatic cancer cells expressing mutant KRAS^G12D^ via treatment with EVs loaded with siRNA that targets KRAS^G12D^. EVs were derived from human mesenchymal stromal cells that we isolated from Wharton’s jelly of umbilical cords and cultured. Furthermore, the use of UC-MSC-derived EVs as a drug delivery system for doxorubicin (DOXO) was tested with pancreatic cancer cells in vitro.

## 2. Materials and Methods

### 2.1. Biological Samples and Culture of Mesenchymal Stromal Cells

This study was approved by the ethics committee (CE2895) of Jules Bordet Institute (Bruxelles, Belgium). Human umbilical cords were collected after full-term deliveries after written informed consent was obtained from the mothers. Segments (5 to 10 cm) were cut and stored at room temperature in sterile phosphate-buffered saline (PBS) supplemented with penicillin/streptomycin until they were used in the laboratory (within 24 h).

An MSC primoculture was obtained according to an explant method developed in our laboratory [16]. Briefly, the umbilical cord segments were longitudinally sectioned to expose Wharton’s jelly (WJ). Some incisions were made in the matrix with a sterile scalpel to allow a larger area of the tissue to contact the plastic surface. The cord segments were then transferred to a 10 cm^2^ Petri dish and cultured for 5 days in Dulbecco’s modified Eagle medium (DMEM) (Lonza, Verviers, Belgium) supplemented with 15% fetal bovine serum (FBS) (Sigma Aldrich, Saint-Louis, MO, USA), 2 mM L-glutamine (Lonza, Verviers, Belgium) and 1% penicillin-streptomycin-amphotericin B mixture (Lonza, Verviers, Belgium). The cultures were maintained in a humidified atmosphere with 5% CO_2_ at 37 °C. After 5 days, the cord segments were discarded, and the medium was replaced. The cells were then grown until they reached subconfluence (80–90%). The medium was changed every week. WJ-MSCs were expanded until passage 5. WJ-MSCs were harvested after detachment by incubation with TrypLE Select solution (Gibco) for 10 min and counted, and then, their phenotype and differentiation capacities were analyzed to confirm MSC identity. Adipogenic, osteogenic and chondrogenic differentiation was induced in adapted media (NH media, Miltenyi Biotec) according to the manufacturer’s instructions and as previously described by our group [17].

### 2.2. Cell Lines

Three cell lines were purchased from American Type Culture Collection (ATCC, Manassas, VA, USA). Two pancreatic cell lines, namely, PANC-1 and BxPC-3, expressed mutant KRAS^G12D^ and wild-type KRAS, respectively, and the colon cell line, LS180, also expressed mutant KRAS^G12D^. All the cell lines were cultured in DMEM that was supplemented as previously described.

### 2.3. Phenotypic Analysis of WJ-MSCs

Cells were harvested after detachment with TrypLE Select, washed in PBS (Miltenyi Biotec, Bergisch Gladbach, Germany) and incubated for 30 min with the following monoclonal antibodies: CD105-FITC (Ancell corporation, Bayport, NY, USA), CD73-PE (Miltenyi Biotec, Bergisch Gladbach, Germany), CD146-PC5 (Beckman Coulter, Analis, Suarlée, Belgium), CD166-PE (BD Biosciences, Erembodegem, Belgium), CD45-PC7 (BD Biosciences, Erembodegem, Belgium), HLA-ABC- PC5 (BioLegend, San Diego, CA, USA), HLA-DR-PC5 (Beckman Coulter, Analis, Suarlée, Belgium), CD47-APC (Miltenyi Biotec, Bergisch Gladbach, Germany) and CD200-PC7 (BD). After washing with PBS, the cells were fixed with 8% formaldehyde. Data were acquired and analyzed on a MacsQuant Analyzer (Miltenyi Biotec, Bergisch Gladbach, Germany).

### 2.4. EV Isolation and Characterization

To isolate EVs, WJ-MSCs at subconfluence were cultured for 24 h in medium without serum (FBS) to avoid contamination with FBS-derived vesicles, as described by Crompot et al. [18]. Serum deprivation for longer than 24 h did not induce WJ-MSC senescence or morphological modifications. We used one liter (L) of WJ-MSC supernatant to purify EVs. Cell-free supernatants were obtained by centrifugation at 300× *g* for 5 min. The supernatants of WJ-MSC cultures were then concentrated with a 3 kDa Macrosep advance centrifugal device (Pall Life Science, New York, NY, USA). The supernatants were then subjected to centrifugation at 150,000× *g* for 1 h at 4 °C (Ultracentrifuge MX 120+, Swinging Bucket rotor S50-ST, k-factor 77, Thermo Fisher Scientific, Waltham, MA, USA). An alternative protocol, summarized in Figure 1, allowed us to skip the Macrosep concentration step. Indeed, the larger capacity ultracentrifuge that could process a larger volume allowed the processing of more than 500 mL per run (Ultracentrifuge Sorvall WX+ 80, Fixed-Angle rotor Fiberlite F37L-8x100, k-factor 168, Thermo Fisher Scientific, Waltham, MA, USA). The pellets were resuspended in 200 μL of PBS (Lonza, Verviers, Belgium) and stored at −20 °C until use.

After purification, the EVs were identified and characterized. First, the morphology of WJ-MSC EVs was evaluated by transmission electron microscopy (TEM) as previously described [18]. The EV count and size repartition were measured by Nanosight technology (Nanosight Ltd., Milton Park, UK). Following the standard characterization guidelines of ISEV [19], EV phenotypes were characterized by the latex bead flow cytometry technique. The expression of tetraspanins, MSC and hematopoietic cell markers and CD47 (“don’t eat me” signal) was analyzed.

Aldehyde/sulfate latex beads (10%, 4% *w*/*v*, 4 μm) (Thermo Fisher Scientific, Waltham, MA, USA) were incubated with 10^8^ EVs for 1 h at room temperature. Thereafter, free binding sites on the beads were blocked with glycine buffer (100 mM). The beads were washed with PBS and centrifuged for 5 min at 300× *g*. Conjugated monoclonal antibodies were then incubated with EV-coated beads for 30 min at room temperature. These markers were used to detect tetraspanins: CD63-PE (Miltenyi Biotec, Bergisch Gladbach, Germany), CD9-FITC (BD) and CD81-PC7 (BioLegend, San Diego, CA, USA). We also used markers described in the phenotypic analysis of UC-MSCs to evaluate the phenotype of EVs.

### 2.5. Uptake of EVs by Cell Lines

EVs were labeled with PKH67, a green fluorescent dye that labels lipid membranes (Sigma Aldrich, Saint Louis, MO, USA), according to the manufacturer’s protocol, and EV uptake by PANC-1, BxPC-3 and LS180 cells was evaluated by flow cytometry (MACSQuant Analyzer). Briefly, EVs were incubated with 2 μL of PKH67 (2 μM) for 5 min, incubated in 1% bovine serum albumin (BSA) to stop the reaction, washed with PBS and centrifuged at 150,000× *g* to remove excess dye. Unlabeled EVs and PBS incubated with PKH67 were used as negative controls. We determined the kinetic uptake of EVs by 50,000 PANC-1/BxPC-3/LS180 cells at 0, 30 min, 1 h, 2 h, 3 h and 24 h.

### 2.6. Loading of EVs with siRNA

Electroporation is a simple and quick method. It consists of creating temporary small pores in the EV membrane under the action of an electric field, which increases membrane permeability. Molecules enter the EVs through diffusion, and the membrane integrity quickly recovers after drug loading.

EVs were electroporated using a single 4 mm cuvette and a Gene Pulser Xcell Electroporator (Bio-Rad, Hercules, CA, USA) to load the EVs with siRNAs. The total volume of electroporation buffer (Bio-Rad, Hercules, CA, USA) was 400 μL. Electroporation was performed at 400 V and 125 μF, and the cuvette was transferred to ice. The EVs were washed with PBS and resuspended in PBS before being used in biological assays. Since the EV concentration can affect electroporation efficiency, 2.10^9^ EVs and 1 μg of siRNA were mixed in 400 μL of electroporation buffer (Bio-Rad, Hercules, CA, USA). Different types of siRNA were tested: scrambled (control) siRNA, KRAS^G12D^ siRNA, unconjugated siRNA or siRNA conjugated to AlexaFluor 647 (Qiagen, Antwerpen, Belgium) to verify siRNA delivery. EVs + siRNA without electric pulses were considered negative controls in the biological assays.

The target sequence was as follows: 5′-AAGTTGGAGCTGATGGCGTAG-3′.

The KRAS^G12D^ siRNA sequences were as follows: sense strand: 5′-GUUGGAGCUGAUGGCGUAGTT-3′ and antisense strand: 5′-CUACGCCAUCAGCUCCAACTT-3′. The scramble siRNA sequence was designed by the supplier and delivered under the product name AllStars Negative control siRNA (Qiagen, Antwerpen, Belgium).

### 2.7. Drug Encapsulation in EVs

In this study, DOXO was chosen as a model drug because of its natural fluorescence. After excitation with a 470 nm laser, DOXO emits a signal at 595 nm, and we could therefore follow its kinetics of loading and release by flow cytometry and fluorimetry. The loading of DOXO (Sigma, Saint-Louis, MO, USA) into EVs was performed by loading MSCs with the drug before EV isolation. In this endogenous production approach, MSCs were incubated with different concentrations of doxorubicin (10 and 50 μM) for 24 h in FBS-free medium. Drug-loaded EVs in the conditioned medium were harvested by ultracentrifugation as previously described.

We analyzed fluorescent EVs by the latex bead method and flow cytometry to demonstrate the loading of DOXO into the EVs. The amount of DOXO loaded into EVs was calculated based on a standard curve obtained from a Fluostar Optima microplate reader by determining the absorbance value at a wavelength of 490 nm. The efficiency of encapsulation was calculated by the following formula total DOXO loaded in EVs/total DOXO initially added.

### 2.8. Real-Time PCR Analyses

The expression level of KRAS^G12D^ in cell lines that were treated or not with scramble or KRAS^G12D^ siRNA was evaluated by RT–PCR. After culturing the cell lines (PANC-1, BxPC-3 and LS180) with loaded EVs, the cells were collected, and total RNA was extracted by TRIzol (Tripure isolation reagent, Roche, Mannheim, Germany) and stored at −20 °C until isolation. Complementary DNA (cDNA) was generated from 500 ng of RNA using qScript cDNA SuperMix (Quanta Biosciences, Beverly, MA, USA) according to the manufacturer’s protocol. KRAS^G12D^ mRNA expression was quantified by real-time PCR using Power SYBR Green PCR Master Mix (Life Technologies, Woolston Warrington, UK). Gene expression was normalized to GAPDH gene expression, as an endogenous control, and calibrated by subtracting 10 (chosen arbitrarily) from the ΔCt. The comparative ΔΔCt method was then used for data analysis, and the fold changes in expression were subsequently calculated (fold change = 2^−ΔΔCt^). The primers that were used are described below in Table 1.

### 2.9. Biological Assays

#### 2.9.1. Cell Viability

Loaded EVs were added to cell culture (PANC-1, BxPC-3 and LS180 cells), and the effects of these EVs were evaluated in several biological assays. Cell viability was analyzed using the CellTiter 96^®^ AQueous One Solution Cell Proliferation Assay (MTS assay; Promega, Madison, WI, USA) according to the manufacturer’s instructions. Cells that were treated or not with EVs were added to 96-well plates (5000 cells per well) in triplicate and incubated for 48 h. MTS reagent was added at 20 μL/well and incubated for 2 h. The optical density was measured at 490 nm using a microplate reader (Fluostar Optima, BMG LabTech, Ortenberg, Germany).

#### 2.9.2. Clonogenic Growth Assay (CFU-F)

Cell lines that were treated or not with EVs were seeded in 6-well plates at a density of 500 cells/well. The cells were cultivated in growth media for 14 days and stained with May Grünwald–Giemsa solution to assess colony growth. A colony was defined as a cluster of at least 50 cells.

#### 2.9.3. Transwell Migration

Transwell chambers with 8.0 μm pore membranes in a 12-well format (Corning, NY, USA) were used according to the manufacturer’s protocol. A total of 25,000 cells were seeded per well in the upper chamber in 100 μL of serum-free medium, and 600 μL of complete medium was simultaneously added as a chemoattractant to the lower chamber. After 24 h of incubation at 37 °C, the cells that remained on the upper surface of the membrane were removed with cotton swabs, and the cells on the lower surface of the membrane were considered as cells that had migrated. After staining with 4% paraformaldehyde and 0.1% crystal violet solution, the cells that had passed through the filter were photographed with an inverted microscope (Nikon TMS, Leuven, Belgium).

#### 2.9.4. Flow Cytometric Analysis of Apoptosis

The level of phosphatidylserines on the cell membrane (an indicator of apoptotic cells) was analyzed using Annexin V-FITC (BD) and 7-aminoactinomycin D (7-AAD) as previously described [18]. After 10 min of labeling, the cells were collected, washed in Annexin V binding buffer and analyzed by flow cytometry.

#### 2.9.5. Cell Cycle Analysis

The cell cycle was evaluated by a standard propidium iodide (PI) staining protocol. Cells that needed to be analyzed were collected, washed, permeabilized and incubated with solution containing PI and RNAse (Coulter DNA-Prep Reagent, Hialeah, FL, USA). Tubes were placed at 4 °C in the dark and incubated overnight before analysis by flow cytometry to identify the cell cycle phases. The cells were gated using forward light scatter and side light scatter to exclude cell debris and aggregates from the data. The PI fluorescence of individual nuclei was measured using a MACSQuant flow cytometer, and at least 2 × 10^4^ cells in each sample were analyzed by FCS 4 Express Flow software (De Novo).

#### 2.9.6. Statistical Analyses

The Wilcoxon matched-pairs signed-rank test was used to analyze the statistical significance of differences in the experimental results. *p* < 0.05 was considered to indicate statistical significance. The data were analyzed, and graphics were generated using GraphPad Prism 9.5.1 (GraphPad Software, San Diego, CA, USA).

## 3. Results

### 3.1. Isolation of MSCs from Umbilical Cord (hUC-MSCs)

hUC-MSCs were isolated from umbilical cords as previously described [16] (Figure 1a). A segment of the umbilical cord was seeded in a Petri dish and incubated for 5 days in DMEM supplemented with 15% FBS and antibiotics. Then, the segment was discarded, and the UC-MSCs that adhered to the plastic surface were washed with PBS. At that time, several colonies of cells with fibroblastic morphology were observed. Later, this primoculture was expanded in T75 and T175 flasks until the cells reached subconfluence. By flow cytometry (Figure 2), we observed that the isolated cells were positive for the tetraspanins CD9, CD81 and CD63, as well as the MSC markers CD73, CD166, CD146, CD105, CD200, HLA-ABC and CD47 (“don’t eat me” signal), and negative for the hematopoietic markers CD45 and HLA-DR. CD200 is differentially expressed on MSCs depending on the tissue of origin. Among different sources, hUC-MSCs have the greatest proportion of cells that express CD200 [20].

### 3.2. EV Isolation by Differential Centrifugation and Characterization

The conditioned media were subjected to successive centrifugation and ultracentrifugation to isolate EVs as previously described (Figure 1b). After washing with PBS, the EV pellets were collected.

These EVs met the minimal experimental ISEV criteria. EVs were characterized by TEM, and furthermore, their typical cup-shaped morphology and spherical lipid bilayer vesicle structure, which define EVs, were observed (Figure 3a). NTA technology was used to determine the size of the EVs (Figure 3b). The EV preparations had an approximate average mean size of 191.6 ± 11.96 nm with an average mode of 137.6 ± 7.73 nm (*n* = 5). The EV yields were 9.85 × 10^11^ ± 9.99 × 10^10^ EVs/mL or 8.34 × 10^4^ ± 3.68 × 10^4^ EVs/MSC (*n* = 3).

Via flow cytometry and the latex bead method, we confirmed that EVs express CD105, CD73, CD146, CD146 and CD200, confirming the UC-MSC origin. They also express the tetraspanin markers CD81, CD9 or CD63, which are pan-EV markers, and the “don’t eat me” signal CD47. The EVs were negative for the hematopoietic markers CD45 and HLA-DR, similar to UC-MSCs (Figure 4).

### 3.3. EV Electroporation Leads to siRNA Incorporation

EVs were first electroporated with fluorescent siRNA (scramble-Alexa Fluor and KRAS^G12D^-Alexa Fluor) to monitor their uptake by cell lines at 0, 1, 3 and 18 h (Figure 5a). NTA was performed to confirm that EV characteristics were not altered by electroporation. Substantial uptake by PANC-1 and LS180 cells was observed, and 85 ± 5.6% and 62.5 ± 10% of cells emitted fluorescence after 18 h (*n* = 4, *p* < 0.03), confirming the uptake of siRNA-loaded EVs (Figure 5b). However, the EV uptake by BxPC-3 cells was lower (50 ± 2.2%), probably due to their behavior in culture and their formation of dense colonies, which inhibits EV uptake.

### 3.4. Cellular Uptake of EVs

The PANC-1 and BxPC-3 cell lines (pancreatic cell lines expressing mutant KRAS^G12D^ and wild-type KRAS, respectively) and the LS180 cell line (colon cell line expressing mutant KRAS^G12D^) can take up EVs with a specific kinetic profile, as shown in Figure 6. Histograms display the fluorescence of the PKH67 marker in each cell line at the 0, 1, 3 and 24 h time points.

### 3.5. EVs Loaded with KRAS^G12D^ siRNA Reduce the Expression Level of KRAS^G12D^

By qPCR, we showed that siRNA-loaded EVs can enter cells and are able to release the siRNA as previously described by Kamerkar et al. [13]. KRAS^G12D^ siRNA inhibited KRAS^G12D^ expression, while scramble siRNA exerted no effect (Figure 7). After 120 h of culture, a great majority of PANC-1 and LS180 cells that were treated with KRAS^G12D^ siRNA were dead, while no effect was observed in cells that were treated with the scramble siRNA. Moreover, no significant effect was observed on the BxPC-3 cell line (expressing wild-type KRAS). Indeed, by MTS assay, we observed a significant decrease in viability for PANC-1 (86 ± 1.1% and 49 ± 2.2% of control viability after treatment with CTRL siRNA and KRAS^G12D^ siRNA EVs, respectively, *n* = 5, *p* < 0.05). siKRAS^G12D^ EVs also reduced the viability of LS180 (65 ± 2.6% versus 89 ± 6.8% of control viability, *n* = 3) but had no effect on BxPC-3 (93.3 ± 2.9% versus 90.6 ± 5%, *n* = 3).

### 3.6. The Clonogenicity of PANC-1 Cells Decreases after Treatment with siKRAS^G12D^ EVs

A CFU assay was performed with the PANC-1 cell line (500 cells/well). The cells were treated with EVs at a concentration of 400 EVs/cell. EVs loaded with siKRAS^G12D^ significantly reduced the formation of PANC-1 cell colonies. No change in the CFU-F number was observed after treatment with scramble siRNA-loaded EVs (Figure 8).

### 3.7. siKRAS^G12D^ EVs Decrease the Migration Potential of PANC-1 Cells

A Boyden chamber assay was used to reveal the effect of siKRAS^G12D^ EVs on the migratory behavior of PANC-1 cells (Figure 9). The results of quantification of crystal violet staining at 590 nm demonstrated a decrease in the optical density (OD) in the siKRAS ^G12D^ EV group compared with the siCTRL EV group (31 ± 7% of inhibition, *n* = 5, *p* < 0.05). In addition, microscopic photography confirmed a decrease in the number of labeled cells that crossed through the semipermeable membrane of the Boyden chamber. The results demonstrated that siKRAS^G12D^ EVs efficiently hindered PANC-1 cell migration.

### 3.8. siKRAS^G12D^ EVs Decrease Cell Viability

According to the MTS assay (Figure 10), we observed a significant decrease in viable cell numbers when PANC-1 and LS180 cells (cell lines expressing KRAS^G12D^) were incubated with siKRAS^G12D^ EVs compared to siCTRL EVs. Moreover, no effect was observed in BxPC-3 cells (cell line expressing wild-type KRAS).

### 3.9. siKRAS^G12D^ EVs Induce Apoptosis in KRAS-Mutant Cells

siKRAS^G12D^ EVs induced apoptotic cell death when PANC-1 and LS180 cells were incubated with the EVs for 120 h, as determined by Annexin-V/7-AAD assay (Figure 11). The total apoptotic cell population after the treatment of PANC-1 and LS180 cells with the EVs increased from 3.1 ± 0.6% to 34.0 ± 5.5% (*p* = 0.0156; *n* = 7) and 6.8 ± 1.9% to 29.3 ± 6.3%, respectively (*n* = 2). No induction of apoptosis was observed in BxPC-3 cells.

### 3.10. Arrest of the Cell Cycle and Accumulation of Cells in the S Phase

The results of propidium iodide labeling and flow cytometry analysis revealed cell cycle arrest with the accumulation of cells in the S phase (Figure 12). Importantly, no effect of modified EVs was observed on BxPC-3 cells (cell line expressing wild-type KRAS). In PANC-1 cells, 10.54 ± 3.86% and 21.90 ± 3.5% of the cells were in the S phase after treatment with siCTRL and siKRAS^G12D^ EVs, respectively (*p* < 0.05 *n* = 6). In contrast, the percentage of BxPC-3 cells in the S phase remained stable (21.4 ± 1.60% and 18.3 ± 1.30% (*n* = 2) after treatment with control and siKRAS^G12D^ EVs, respectively).

### 3.11. UC-MSC-Derived EVs as Natural Drug Delivery Vehicles

The potential of EVs to function as drug nanocarriers was evaluated in the second part of this study, and their efficacy was tested in vitro with PANC-1 cells. Doxorubicin (DOXO) was chosen as a model drug in this study. DOXO is naturally fluorescent; therefore, its kinetic loading and release could be monitored by flow cytometry and fluorimetry. An endogenous production method was chosen based on the strategy of endogenous EV production by MSCs so that the EVs were loaded with specific cargo. First, UC-MSC sensitivity to different concentrations of DOXO was evaluated after 24 h by trypan blue exclusion assay. Interestingly, UC-MSCs were resistant to the toxic effects of DOXO. As shown in Figure 13, even at 50 μM, DOXO did not significantly affect cell viability (22 ± 5% cytotoxicity, *n* = 4).

#### 3.11.1. Treatment of UC-MSCs with DOXO

UC-MSCs were incubated with DOXO (10 and 50 μM) for 24 h, allowing the drug to be taken up by the cells and subsequently secreted into a conditioned medium via EVs. The rapid uptake of DOXO by UC-MSCs was observed by flow cytometry (Figure 14a), and more than 58 ± 14% of cells were fluorescent after 3 h of incubation (*n* = 4, *p* < 0.03) (Figure 14b). The decrease in fluorescence at 24 h seemed to correspond to the release of DOXO into the medium (Figure 14a). As shown in Figure 14c, the DOXO levels in CM and in cells were, respectively, 0.89 ± 0.27 and 6.6 ± 0.8 μM for a DOXO cell loading of 10 μM (*p* < 0.01, *n* = 4). After 24h incubation of UC-MSCs with 50 μM DOXO, the drug levels found in CM and cells were, respectively, 11.43 ± 2.9 and 29.54 ± 2.9 μM (*p* < 0.005, *n* = 6). We thus observed in cells 67 ± 8 and 59 ± 6% of the initial DOXO concentration used to treat UC-MSCs.

#### 3.11.2. Characterization of DOXO-Loaded EVs

The EVs produced by UC-MSCs that were treated with DOXO were purified by ultracentrifugation as previously described and characterized. An increase in the EVs produced by UC-MSCs after DOXO treatment (15.6 ± 4.8 × 10^10^/mL for untreated MSCs versus 27 ± 10 × 10^10^/mL for 10 and 50 μM DOXO-treated cells, *n* = 5) was observed by NTA (Figure 15), and the size of EVs produced by DOXO-treated cells was slightly increased (195 ± 18 nm for untreated cells versus 215 ± 33 nm and 217 ± 36 nm for 10 and 50 μM DOXO-treated cells, respectively, *n* = 5). However, these differences were not statistically significant. The EVs produced by DOXO-treated MSCs were fluorescent, as observed by flow cytometry with latex beads (Figure 16a). A stronger fluorescence intensity was observed after loading with a 50 μM drug, confirming the presence of DOXO in EVs. The levels of DOXO in the purified EVs were measured by fluorometric assay. Concentrations of 0.967 ± 0.216 and 4.073 ± 0.862 μM in 10^10^ EVs were observed for EVs derived from MSCs loaded with 10 and 50 μM DOXO, respectively, corresponding to ± 10% of the total DOXO used to treat the MSCs (Figure 16b).

#### 3.11.3. Uptake of DOXO EVs by PANC-1 Cells and Their Anticancer Activity

As shown in Figure 17a, DOXO-loaded EVs were rapidly taken up by PANC-1 cells; the cells emitted fluorescence as early as 1 h posttreatment, and the fluorescence intensity peaked at 5 h. After 48 h, we evaluated the effect of different concentrations of DOXO on PANC-1 cell viability. A dose-dependent effect was observed, and 10 and 100 μM DOXO induced apoptosis in 50 ± 1.7% (*p* < 0.03, *n* = 4) and 59.5 ± 0.3% of the PANC-1 cells (*p* < 0.03, *n* = 4), respectively (Figure 17b). Pancreatic cells treated with DOXO-loaded EVs also underwent apoptosis; 17.8 ± 0.64% (*p* < 0.02, *n* = 8) and 57 ± 2% (*p* < 0.02, *n* = 8) pancreatic cells were apoptotic after treatment with EVs obtained from 10 and 50 μM DOXO-treated MSCs versus 12 ± 0.56% of pancreatic cells after treatment with EVs obtained from untreated cells (Figure 17c). Interestingly, DOXO loaded in EVs was clearly more effective than the free drug at a lower concentration. Indeed, as reported before, the mean DOXO concentrations were 0.967 ± 0.216 μM and 4.073 ± 0.862 μM in 10^10^ EVs derived from MSCs loaded with 10 and 50 μM, respectively.

## 4. Discussion

Specific targeting of mutant KRAS using small interfering RNAs has shown great promise for the treatment of PDAC [21]. Unfortunately, siRNA therapy is limited by low serum half-life, inadequate site-specific delivery, intracellular digestion and transient therapeutic targets [6]. Different compositions of NPs have been tested to target KRAS and its signaling pathway via siRNAs, but none of these NPs have translated to clinical application [7].

The current work demonstrated the feasibility of specifically targeting pancreatic cancer cells with the KRAS^G12D^ mutation via EVs that carry siRNA to target mutant KRAS. The EVs were derived from human MSCs that were isolated and amplified from umbilical cord Wharton’s jelly. Limitations such as the invasive nature of harvest strategies, donor side effects, cell yield and cellular biological characteristics are factors that determine ideal cell sources [22]. The umbilical cord, which is a postnatal tissue that is normally discarded after birth, provides a noninvasive source for harvesting MSCs in high yields [23]. Moreover, UC-MSCs exhibit a high proliferation rate, are hypoimmunogenic (loss of HLA class II) [24] and produce a large number of EVs [25,26].

The ultracentrifugation method, based on the size and density of the EVs, was chosen to isolate EVs. This method is simple, inexpensive, applicable for large volumes of supernatants and produces a good yield. The EVs were characterized by transmission electron microscopy (TEM), nanoparticle tracking analysis (NTA) and latex-bead-based flow cytometry. Positive expression of tetraspanins (CD9, CD63 and CD81), as well as the mesenchymal markers CD105, CD73 and CD166, and negative expression of CD45 and HLA-DR were observed on EVs. EVs express a phenotype similar to the mesenchymal cells from which they are derived. As Kamerkar et al. previously described for exosomes derived from mouse skin fibroblasts, EVs also express CD47, which can interact with SIRPα located on the membranes of macrophages, preventing their phagocytosis. CD47 expression facilitated the in vivo biological activity of EVs [27]. We demonstrated rapid EV internalization by pancreatic cell lines, but each cell line exhibited different kinetics from the others. The rapid internalization of MSC-derived EVs, in vitro and in vivo, was also observed in different cell types, including leukemia cells [18,28], breast cancer cells [29] and glioblastoma cells [30].

By electroporation, we loaded EVs with scramble and KRAS^G12D^ siRNA, and we showed that siRNA-loaded EVs entered cells and delivered the siRNA. Indeed, KRAS^G12D^ siRNA significantly inhibited the expression of KRAS^G12D^, while scramble siRNA had no effect. This inhibitory effect of KRAS^G12D^ siRNA on the expression of KRAS^G12D^ transcripts was comparable to that observed by Kamerkar et al. [13] with mouse skin exosomes but weaker than the effect reported in the study by Mendt et al. [31] using bone marrow MSC-derived EVs. This difference is likely explained by the amount of loaded EVs used to treat the cells.

After 120 h of culture, the majority of KRAS^G12D^-mutant cells (PANC-1 and LS180 cells) treated with KRAS^G12D^ siRNA EVs were dead, whereas no effect was observed after treatment with EVs loaded with scramble siRNA. We observed an impact of KRAS^G12D^ siRNA EVs on cell clonogenicity (CFU-F), migration (Boyden chamber and wound healing assay), viability (MTS assay), apoptosis (Annexin-V/7-AAD) and cell cycle progression (incorporation of propidium iodid). It is important to note that no effect of KRAS^G12D^ siRNA EVs on the wild-type BxPC-3 cell line was observed, confirming the specificity of the treatment. The treatment of mutant cells with KRAS^G12D^ siRNA EVs resulted in an accumulation of cells in the S phase of the cell cycle. Other studies associating NP and KRAS siRNAs also observed strong arrest of treated PANC-1 cells in the S phase [32].

In the study by Mendt et al. [31], EVs derived from bone marrow MSCs and loaded with KRAS^G12D^ siRNA were able to inhibit tumor growth in xenograft mouse models. Surprisingly, after peritoneal injection, EVs preferentially localized in pancreatic tumor tissues according to a mechanism that is not yet elucidated. Increased KRAS-induced macropinocytosis in tumor cells may be a contributing mechanism [33]. Treatment with KRAS^G12D^ siRNA-loaded EVs significantly reduced tumor volume and prolonged survival in several PDAC mouse models [13,31].

EVs are efficient transporters of cellular materials and are natural nanoparticles that do not induce undesirable immunological responses. These properties have enabled their use as drug delivery systems. However, the drug loading capacities and the release of these drugs in target cells strongly depend on the cellular origin of the EVs [34]. In pancreatic cancer, the differences observed between the initial tumor response and very poor long-term survival are the consequences of rapidly acquired chemoresistance; these differences represent a major new research direction in tumor treatment. This chemoresistance is multifactorial and includes chemical instability, low cellular absorption and short half-lives, but the stroma of pancreatic cancer seems to determine the resistance to chemotherapy. Indeed, the fibrous architecture of stromal cells creates a physical barrier through which chemotherapies such as gemcitabine cannot cross [35]. Nanotechnology has emerged as a promising way to improve drug delivery to the pancreas via passive and active targeting mechanisms, allowing us to overcome this pathophysiological barrier.

We evaluated the potential of EVs derived from umbilical cord MSCs to be used as drug nanocarriers, and we tested their efficacy on pancreatic cancer cells in vitro. DOXO was chosen as a model drug in our work because it is naturally fluorescent [36]. DOXO emits signals at 595 nm after excitation with a 470 nm laser, and we can therefore monitor its loading kinetics and release by flow cytometry and fluorimetry. On the other hand, this drug has been frequently investigated for the development of various types of NPs as drug delivery systems [37,38,39]. We chose the endogenous EV production method based on the strategy of endogenous EV production by MSCs so that the EVs could be loaded with a specific cargo. Indeed, MSCs are relatively resistant to chemotherapeutic drugs and have a high degree of absorption, allowing the release of charged EVs. Different mechanisms have been proposed to explain this resistance, including an increase in heat shock protein expression, the effects of tubulin and the inhibition of apoptosis by the overexpression of antiapoptotic factors such as Bcl2 and Bcl-xL [40]. The overexpression of the MIF (macrophage migration inhibitory factor) facilitates bone-marrow-derived MSC resistance to DOXO via PI3K-Akt pathway activation [41].

UC-MSCs are resistant to the cytotoxic effects of DOXO, and at concentrations of 50 μM, cytotoxic effects of less than 25% were observed after 24 h of treatment. We confirmed by flow cytometry the rapid uptake of DOXO into MSCs, and more than 80% MSCs were fluorescent after 3 h of incubation. EVs produced by DOXO-treated MSCs and isolated by ultracentrifugation were well fluorescent and contained DOXO (0.967 ± 0.216 μM and 4.073 ± 0.862 μM for 10^10^ EVs derived from MSCs treated with 10 and 50 μM DOXO, respectively), corresponding to ±10% of the total amount of DOXO added to the cells. Bagheri et al. electroporated DOXO into EVs derived from mouse bone marrow MSCs and reported a loading efficiency of 17 ± 5% [42]. Wei et al. incubated EVs derived from bone marrow MSCs with DOXO and observed an encapsulation efficiency of ±12% [43]. DOXO-loaded EVs quickly entered PANC-1 cells, and after 48 h, the pancreatic cells underwent apoptosis; this effect was especially notable under loading conditions of 50 μM. Moreover, the DOXO encapsulated in the EVs was clearly more effective than the free drug and at lower concentrations. Following the active loading method, DOXO-loaded EVs exerted significantly greater tumor inhibitory effects and fewer side effects than free DOXO in xenograft mouse models of osteosarcoma [44]. DOXO-loaded EVs increased retinoblastoma cell apoptosis rates and induced caspase 3/7 activation more efficiently than the free drug [45]. EVs derived from CSM and loaded with DOXO were also effective against hepatocellular carcinoma and induced a better inhibitory effect in vitro and in vivo [46]. Another study confirmed the faster uptake of DOXO-loaded EVs by different cell types, allowing high drug accumulation and leading to superior in vitro antitumor activity compared to free drug and liposomal formulations [47].

The loading of paclitaxel and gemcitabine into EVs derived from different cell types has also been reported in the literature. In a murine model of melanoma and in vitro, EVs derived from a monocytic cell line and loaded with paclitaxel by sonication were 50× more cytotoxic to cancer cells than the free drug [48]. Recently, MSCs isolated from the gingiva were loaded with paclitaxel and showed good resistance to this drug up to 10 μg/mL, but an increase in the numbers of cells in the G2/M phase was observed, consistent with the mode of action of paclitaxel [49]. The treatment of MSCs with paclitaxel allowed the production of drug-loaded EVs without altering their morphology, and these EVs were capable of inhibiting the growth of pancreatic and squamous cell carcinoma cells in vitro. Paclitaxel was also loaded into EVs derived from umbilical cord MSCs, and these EVs induced apoptosis and reduced the levels of the epithelial–mesenchymal transition-related proteins in HeLa cervical cancer cells [50].

Recently, dental-pulp-derived MSCs were treated with gemcitabine, and the loaded EVs were purified by tangential flow filtration. These loaded EVs inhibited the growth of the pancreatic carcinoma lines PANC-1 and MiaPaca-2 in vitro. Although a small amount of gemcitabine was encapsulated in the EVs (0.7812 μg/mL), the inhibitory activity was similar to that of a 2 μg/mL free drug [51]. Interestingly, bone marrow MSC-derived EVs were coloaded with gemcitabine and paclitaxel by electroporation and sonication. These loaded EVs showed good uptake by pancreatic cancer cells in vitro and in vivo, as well as very good antitumor efficacy [52]. These results suggest that the administration of drug-coloaded EVs could overcome the obstacles in treating pancreatic cancer, such as chemoresistance and the pathological barrier, and would represent a promising strategy for targeted therapy in pancreatic cancer.

## 5. Conclusions

In conclusion, we were able to demonstrate the feasibility of using EVs derived from umbilical cord MSCs to transport siRNAs that target the KRAS^G12D^ mutation or a drug into pancreatic cancer cells. EVs derived from umbilical cord MSCs represent an excellent biological nanovehicle for the delivery of siRNAs or drugs for the treatment of pancreatic cancer. The coloading of small interfering RNAs with drugs could also be considered as previously described. Indeed, the coloading of DOXO and a siRNA targeting the KRAS^G12D^ mutation has already been described in the literature but in nanocomposites based on polymers and gold nanorods. A synergistic effect of DOXO and siRNA KRAS^G12D^ inhibited tumor growth by more than 90% in xenograft mouse models of pancreatic cancer [32].

Other targets could be considered in the context of pancreatic cancer. It would be interesting to evaluate the combination of agents that target components of the microenvironment, such as carcinoma-associated fibroblasts (CAFs), with conventional chemotherapies or immunotherapies in pancreatic cancer. Nanomedicines could be developed to increase the immunogenicity of pancreatic cancer cells, inactivate CAFs, increase the antigen-presenting capacity of dendritic cells, counteract the immunosuppression of the tumor microenvironment and improve the infiltration of cytotoxic T cells. For example, the antitumor efficacy of nanoparticles loaded with irinotecan, a synthetic derivative of camptothecin, was synergistically increased by anti-PD1 in an orthotopic model of pancreatic cancer [53].

Although the results from in vitro and preclinical animal studies have been encouraging, different steps are still needed to increase quality control and procedure standardization. Indeed, different protocols for the purification, quantification and characterization of EVs are still in use [54]. The lack of standardized methods for the isolation and purification of EVs, the limited efficiency of drug encapsulation in EVs, the isolation of EVs contaminated with impurities (cellular debris, nonexosomal vesicles, proteins) and insufficient production yield remain major challenges. For clinical applications, the evaluation of the storage conditions, pharmacokinetics and biodistribution of loaded EVs is needed. In addition, the culture of MSCs that produce EVs must also be considered for large-scale production. Bioreactors for cell expansion must provide enough EVs for clinical-grade production [55].

## Figures and Tables

**Figure 1 cancers-15-02901-f001:**
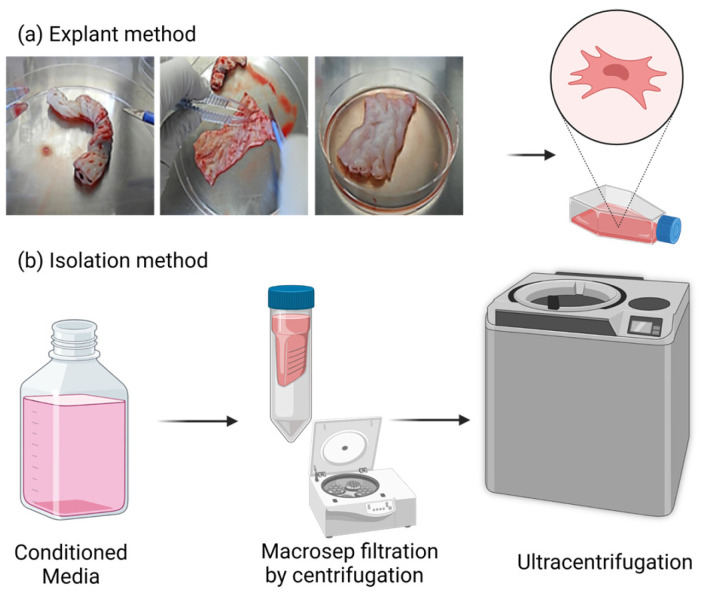
UC-MSC isolation by the explant method and isolation of EVs by ultracentrifugation. (**a**) Segments of umbilical cord were cut and seeded in Petri dishes, and MSC primocultures were isolated and expanded in T75 and T175 flasks. (**b**) EV isolation from conditioned media by differential centrifugation.

**Figure 2 cancers-15-02901-f002:**
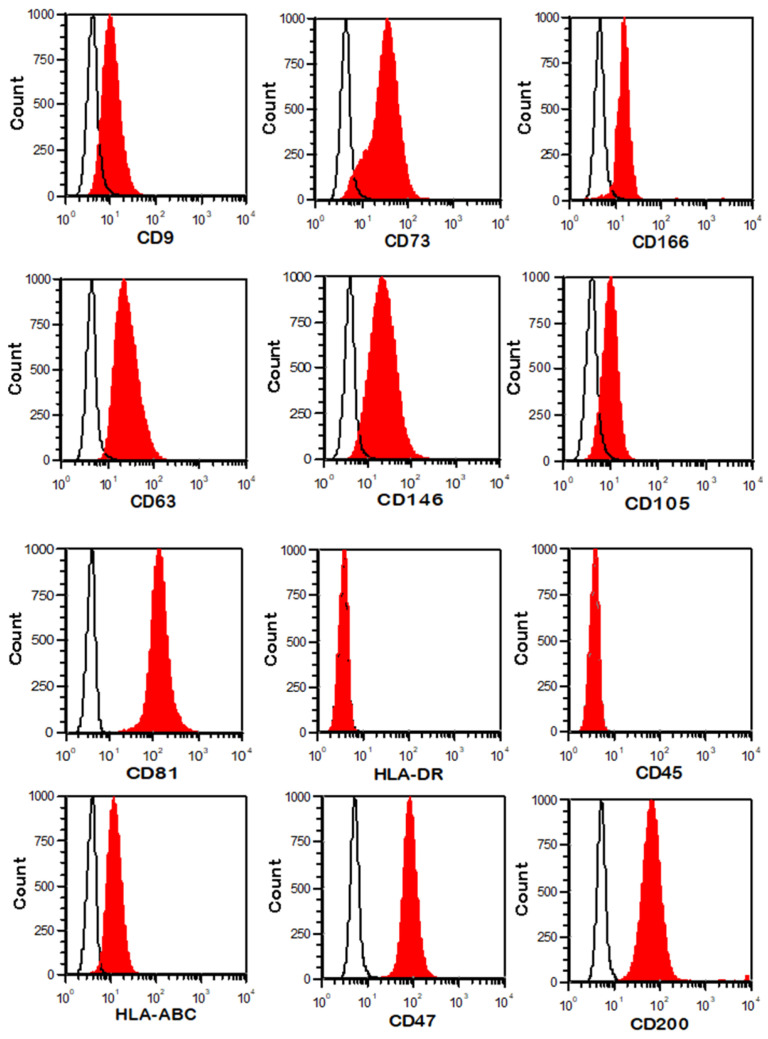
Phenotypical characterization of UC-MSCs by flow cytometry (unfilled histogram = CTRL/filled histogram = labeled cells). Histograms show positivity for the tetraspanins CD9, CD63 and CD81, as well as the MSC markers CD73, CD166, CD146, CD105, HLA-ABC, CD47 and CD200, and negativity for the hematopoietic markers HLA-DR and CD45.

**Figure 3 cancers-15-02901-f003:**
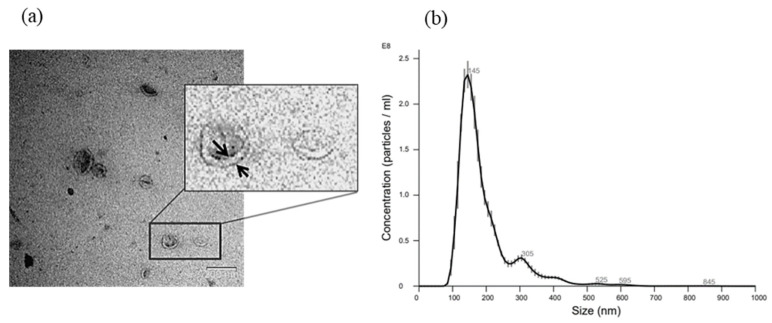
(**a**) EV characterization using transmission electronic microscopy (TEM): Spherical lipid bilayer vesicles presenting typical cup-shaped morphology. EV characterization by nanotracking analysis. (**b**) Representative histogram of EV size distribution and amount of EVs produced: meaning size: 192.1 ± 11.96 nm and size mode: 138.1 ± 7.73 nm.

**Figure 4 cancers-15-02901-f004:**
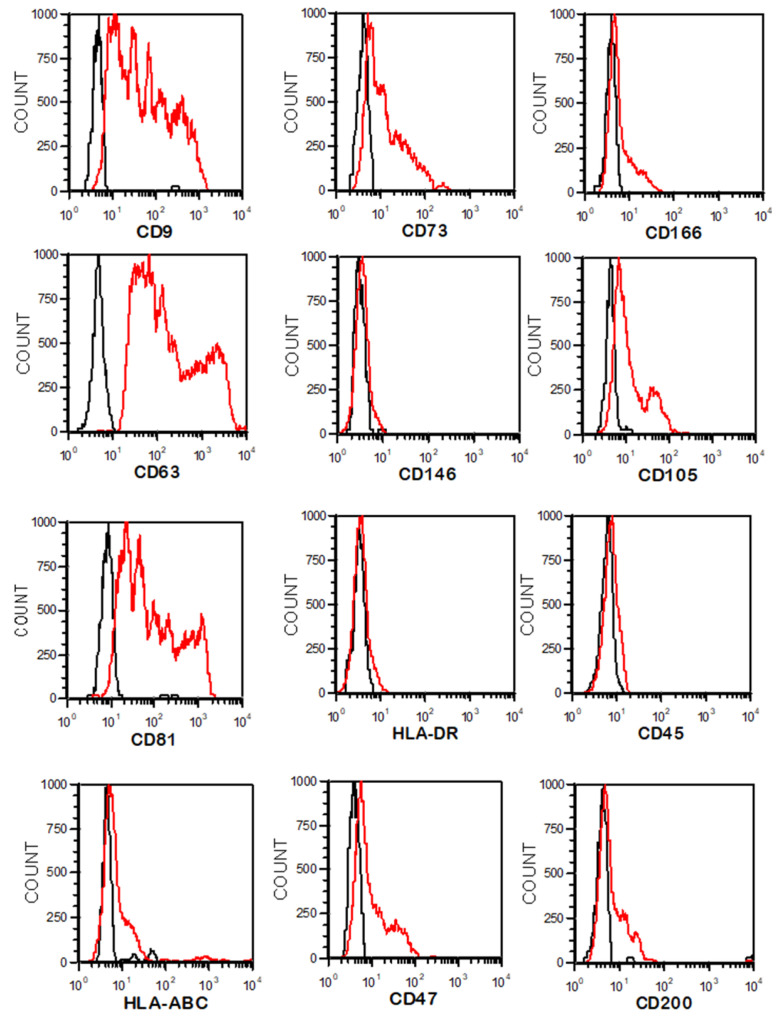
Phenotypical characterization of EVs derived from hUC-MSCs by flow cytometry (black line = CTRL, red line = labeled cells). Histograms show positivity for the tetraspanins CD9, CD63 and CD81, as well as for the MSC markers CD73, CD166, CD146, CD105, HLA-ABC, CD47 and CD200, and negativity for the hematopoietic markers HLA-DR and CD45.

**Figure 5 cancers-15-02901-f005:**
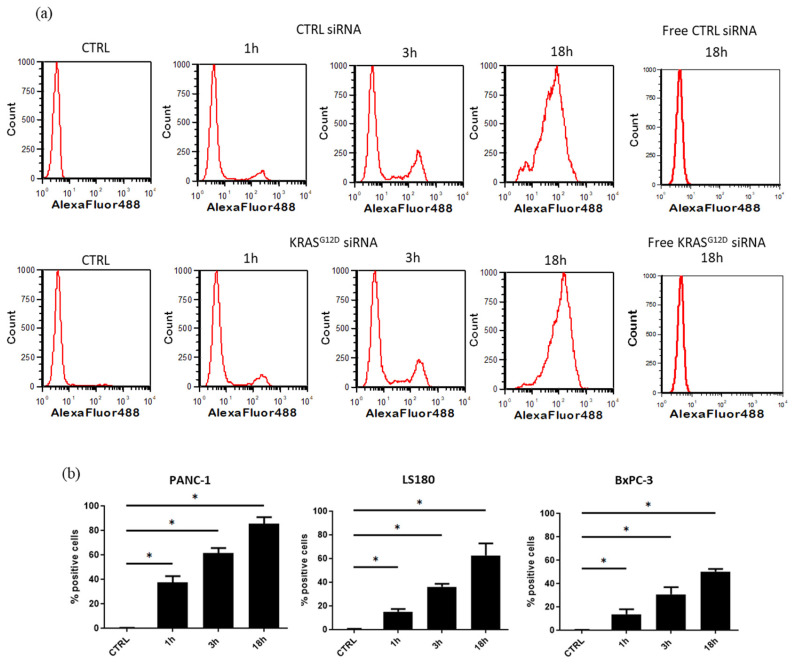
(**a**) Kinetics of fluorescent siRNA-loaded EV uptake by PANC-1 cells at 0, 1, 3 and 18 h and free siRNA -CTRL and -KRAS^G12D^ at 18 h. (**b**) Kinetics of fluorescent siRNA KRAS^G12D^ loaded EVs in PANC-1, LS180 and BxPC3 (*n* = 4, * *p* < 0.03).

**Figure 6 cancers-15-02901-f006:**
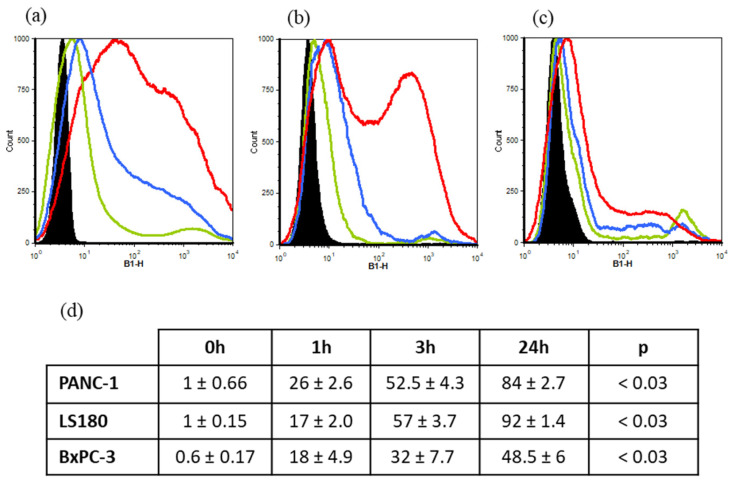
EV uptake by PANC-1 (**a**), LS180 (**b**) and BxPC-3 (**c**) cells monitored after the PKH67 staining of EVs at 0 h (black), 1 h (green), 3 h (blue) and 24 h (red). Each cell line displayed a specific kinetic behavior. A total of 5000 cells were incubated with PKH67-labeled EVs (400 EVs/cells). (**d**) EV uptake by cell lines expressed in mean percentage ± SEM of PKH67 positive cells (*n* = 4).

**Figure 7 cancers-15-02901-f007:**
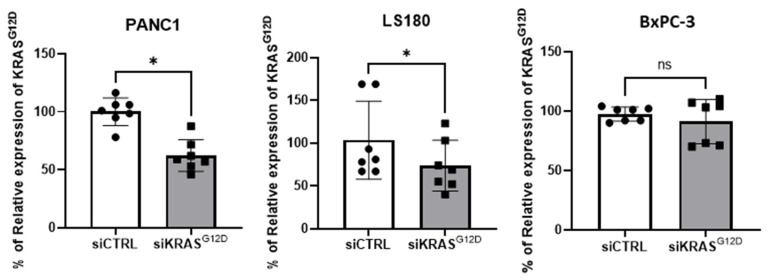
Significant decrease in KRAS^G12D^ expression in the PANC-1 and LS180 cells lines that express the mutant KRAS^G12D^ protein after treatment with scramble (where the dot represents individual values) or KRAS^G12D^ siRNA-loaded EVs (where the square represents individual value). No significant decrease was observed in the BxPC-3 cell line that expresses wild-type KRAS^G12D^. (* *p* = 0.0156, *n* = 7).

**Figure 8 cancers-15-02901-f008:**
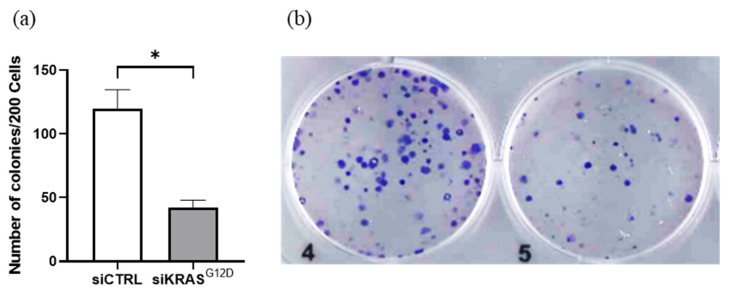
(**a**) Biological effect of EVs loaded with siKRAS^G12D^ on PANC-1 clonogenicity as evaluated by the CFU-F assay. (**b**) Representative image of CFU-F formed by PANC-1 cells after treatment with (4) siCTRL and (5) siKRAS^G12D^. (* *p* = 0.0313, *n* = 6).

**Figure 9 cancers-15-02901-f009:**
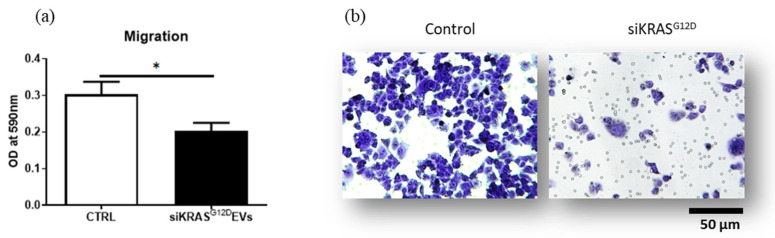
(**a**) siKRAS^G12D^ EVs reduced the migration of the PANC-1 pancreatic cell line. Histograms display the optical density of stained cells that were removed from the Boyden chamber after culture (* *p* < 0.05, *n* = 5). (**b**) The images show decreased migration when PANC-1 cells were incubated with siKRAS^G12D^ EVs in comparison with control conditions.

**Figure 10 cancers-15-02901-f010:**
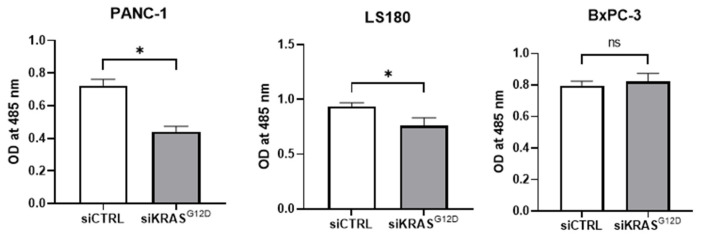
Significant decrease in the viability of PANC-1 and LS180 cells treated with siKRAS^G12D^ EVs and no decrease in the BxPC-3 cell line (expressing wild-type KRAS). (* *p* < 0.04; *n* = 6).

**Figure 11 cancers-15-02901-f011:**
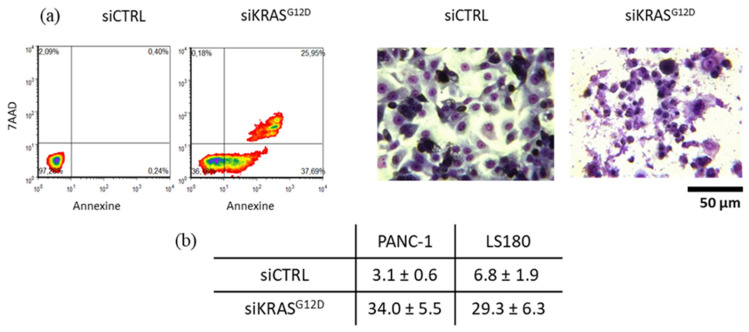
(**a**) Representative flow cytometry analysis of Annexin-V- and 7-AAD staining showing apoptotic and dead PANC-1 cells after treatment with siCTRL and siKRAS^G12D^ EVs and images of May Grünwald–Giemsa-stained cultured cells (20× magnification). (**b**) Table summarizing the percentage of total apoptotic PANC-1 and LS180 cells.

**Figure 12 cancers-15-02901-f012:**
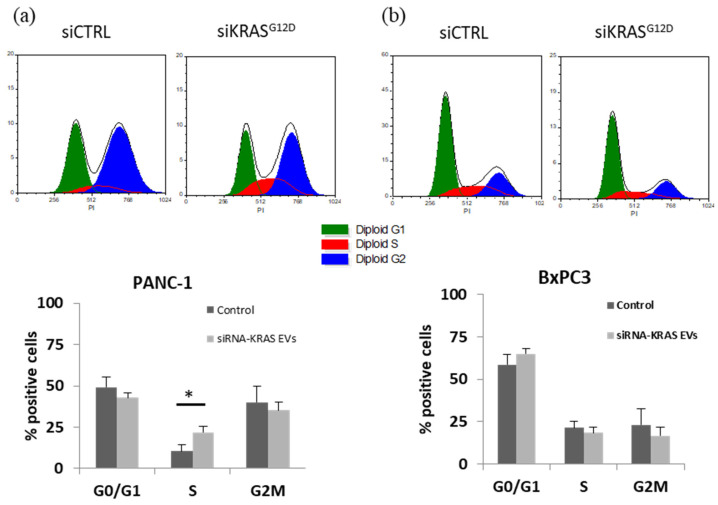
(**a**) Phase S accumulation of PANC-1 cells treated with siKRAS^G12D^ EVs and (**b**) no effect on BxPC-3 cells (* *p* < 0.05 *n* = 6).

**Figure 13 cancers-15-02901-f013:**
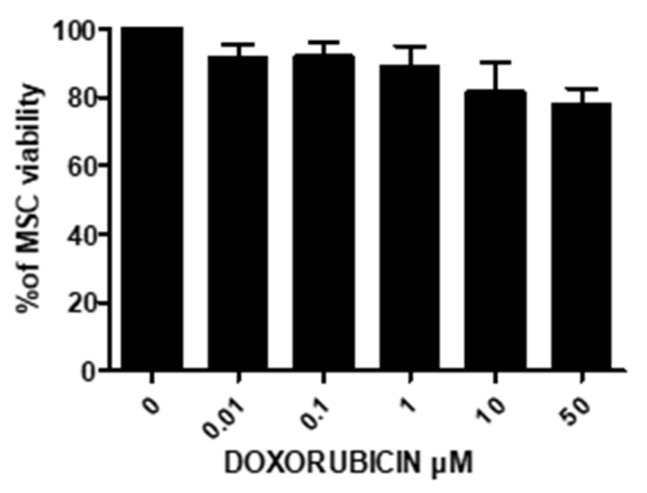
Analysis of UC-MSC sensitivity to doxorubicin (DOXO). The cytotoxicity was evaluated after 24 h by trypan blue exclusion (*n* = 4).

**Figure 14 cancers-15-02901-f014:**
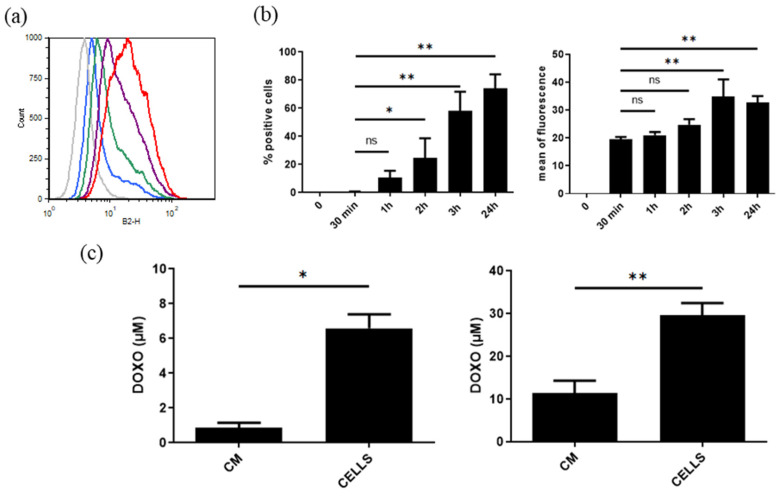
(**a**) Representative kinetics of DOXO uptake by UC-MSCs. DOXO (10 μM) was incubated with UC-MSCs, and after 30 min (grey), 1 h (blue), 2 h (green), 3 h (purple) and 24 h (red), the fluorescence of the cells was evaluated by flow cytometry. (**b**) Mean kinetics of DOXO uptake by UC-MSCs (*n* = 4, * *p* < 0.05, ** *p* < 0.03). (**c**) The levels of DOXO were quantified by fluorimetry (excitation and emission wavelengths were 480 nm and 594 nm, respectively) in cells and in conditioned medium (CM) after 24 h of treatment with 10 μM (*n* = 4, * *p* < 0.01) and 50 μM (*n* = 6, ** *p* < 0.005) of DOXO.

**Figure 15 cancers-15-02901-f015:**
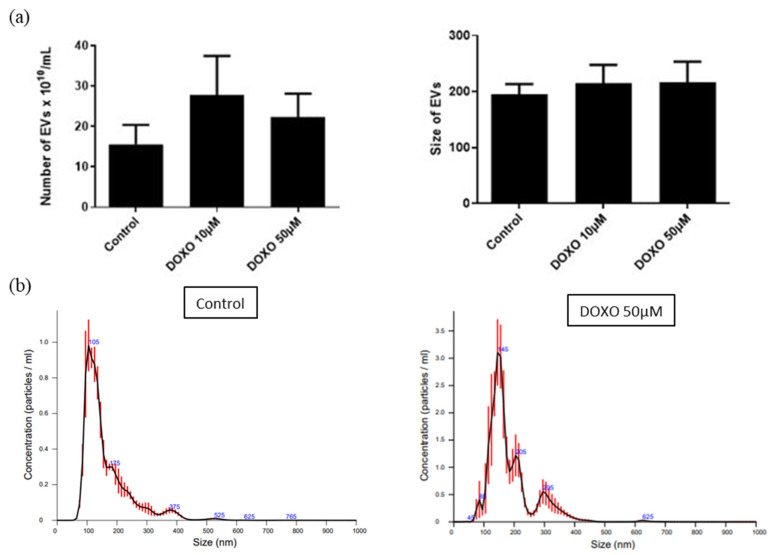
(**a**) NTA analysis of EVs isolated from 10 and 50 μM DOXO-treated UC-MSCs after 24 h (*n* = 5). (**b**) Representative NTA profile of EVs isolated from untreated and DOXO-treated UC-MSCs.

**Figure 16 cancers-15-02901-f016:**
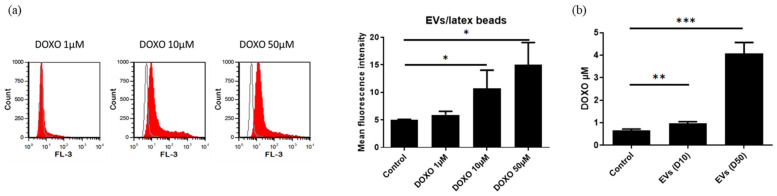
(**a**) Detection of fluorescent EVs produced by 10 and 50 μM DOXO-treated UC-MSCs. The results are expressed as the mean fluorescence intensity (*n* = 4, * *p* < 0.04). EVs were captured on aldehyde/sulfate latex beads and analyzed by flow cytometry. (**b**) DOXO levels in EVs isolated from untreated MSCs and 10 and 50 μM DOXO-treated MSCs (*n* = 7, ** *p* < 0.02, *** *p* < 0.005).

**Figure 17 cancers-15-02901-f017:**
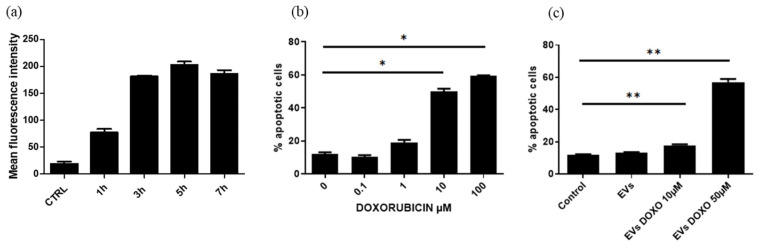
(**a**) Uptake kinetics of DOXO-loaded EVs by PANC-1 cells (*n* = 2). (**b**) Dose-dependent response of PANC-1 cells to DOXO in terms of apoptosis evaluated by Annexin-V/7-AAD staining after 48 h of treatment (* *p* < 0.03, *n* = 4). (**c**) Apoptotic effect of EVs loaded or not with DOXO by treatment of UC-MSCs with 10 and 50 μM DOXO (** *p* < 0.02, *n* = 8).

**Table 1 cancers-15-02901-t001:** KRAS primer sequences.

	Reverse	Forward
KRAS^WT^	TGTAGGAATCCTCTATTGTTGGATCA	AAGAGTGCCTTGACGATACAGCTA
KRAS^G12D^	TTGGATCATATTCGTCCACAA	ACTTGTGGTAGTTGGEGCAGA

## Data Availability

Not applicable.

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
