# Peer review of "Extracellular Vesicles Derived from Human Umbilical Cord Mesenchymal Stromal Cells as an Efficient Nanocarrier to Deliver siRNA or Drug to Pancreatic Cancer Cells"

_cancers, 2023, doi:10.3390/cancers15112901_

Round 1

Reviewer 1 Report

The research article entitled “Extracellular Vesicles derived from human umbilical cord mesenchymal stromal cells as an efficient nanocarrier to deliver siRNA or drug in pancreatic cancer cells” by Draguet et al. present interesting research data envisioning the future application of EVs as therapeutic nanocarrier. The article is well written and easy to follow. However, some data appears to be preliminary and following points need to be considered to improve the quality of data presentation.

1. In the figures 5, 6, 9, 10, 12, 15, 16, 17 the authors show results without data quantification or statistics. Therefore, the data appears to be quite preliminary. Although the shown results are interesting and might fit into the story line of the article they need further replication and a statistical evaluation.

2. Figure 10 shows the effect of siRNA delivery on the transmigration of PANC-1 cells. The bar diagram suggest a quite small effect (~0.22 vs. ~0.18), while in the provided microscopy image the effect appears to be much stronger. It would be more convincing to provide an image that matches the diagram.

3. The authors loaded the EVs with siRNA by electroporation. Did the authors verify that electroporation did not change the shape and characteristics of the EVs e.g. by TEM or NTA?

4. The uptake of EV encapsulated siRNA into PANC-1 cells is shown in figure 5. Why did the authors not show the uptake of EVs into BxPC-3 cells? Is the control (CTRL) in figure 5 free, non EV-encapsulated fluorescent siRNA? If not, the authors need to perform this important control to underpin the role of EVs for the delivery of the siRNA.

5. Similar to the previous point. Did the author show the effect of free DOXO on PANC-1 cells (figure 18)?

Minor comments

-          In line 149: What does the sentence “Other markers were previously described.” mean?

-          In line 231: Please change “... considered cells that had migrated.” to “... considered as cells that had migrated.”

-          In line 262: The word “Obtention” might be changed to “Isolation”.

-          In line 389: The authors mentioned that BxPC-3 cells express wild-type KRAS. It would be good to give this information earlier e.g. in line 336.

-          E.g. in line 447. The authors used a comma as decimal separator instead of a dot. Please change also in the following sections.

Reviewer 2 Report

It was a pleasure to review this manuscript. The study address an important issue of PDAC.

-Introduction should provide a background and history on the use of EVs in PDAC. Additionally, the authors should state clearly their hypothesis and the gap in knowledge that they are trying to address specifically.

-Ensure that (p-value) is mentioned for all the figures even when not statistically significant (for e.e.g figure 10 and 13). Some of the figures are missing the "n" number.

- The quality of figures is very poor. Please provide high resolution images.

Reviewer 3 Report

This manuscript reported a novel extracellular vesicle system which is derived from human umbilical cord mesenchymal stromal cell (UC-MSC) to specifically target pancreatic cancer cells. By loading KRASG12D-targeting siRNA or DOXO, the outstanding targeting ability and drug/siRNA protection ability of this UC-MSC EVs system had proved. However, this article has too many errors and omissions, which is far from the requirement of publication. In summary, I do not recommend this manuscript to be published in Cancers. And I suggest that authors can consider optimizing this paper with the following advice.

1.    For the legend of Fig. 4, the “unfilled/ filled histogram” should be considered for revision. 

2.    At the end of part 3.5, the viability of various cell types should be displayed in the main body, which demonstrated the effectiveness of siKRASG12D. The CCK8 or MTT assay may meet the purpose. 

3.  In Fig. 9, the scale bars have not shown. Besides, screenshots should be avoided used in the paper. (Fig. 9g) 

4.  The scale bars in Fig. 10b have also been missed. And Sample variance analysis by T-test needs to be added to Fig. 10a.

5.   There are several flaws in drawing, such as the unit missing or unmatched in the y-axis (Fig. 15b, 16a), and the figures in paper should be considered drawn by plotting software to avoid rough edge lines and error bars.

6.     In Fig. 18, which two groups are the asterisks represented for?

7.   In flow cytometry analyses in the paper, the number of cells or EVs is far lesser than the number required, which should lift to approximately 10,000.

Round 2

Reviewer 1 Report

The authors addressed most of my comments. However, a few things may still require the attention of the authors.

Comment 1: Although the authors aimed to add data quantifications, number of replicates and statistical evaluations, statistical statements are still absent in some Figures: Figure 5b, 6, 9 and 15. It is not clear to me why the authors added statistics in some figures but not all, although the number of replicates appears to be sufficient (mostly). If authors are not able to provide any statistical evaluation the data sets should be excluded from the manuscript.

Comment 2: Figure 9c and 9d are identical, please show the correct images. Wound healing/scratch assays are not very sensitive and a small number of replicates (n=2) is not enough. Authors should increase the number replicates. Otherwise they should not include these data into the manuscript.

Comment 3: The decimal separator of some numbers is still a comma but should be replaced by a dot. For example, the table shown in Figure 9h.

Comment 4: The sentence “Any modifications of EV characteristics analyzed by NTA were observed after electroporation” (line 335-336) need to be revised. Author may write: “NTA was performed to confirm that EV characteristics were not altered by electroporation”.

Reviewer 2 Report

Thank you for the opportunity to review the submitted manuscript. It is a well conducted study that uses EVs from umbilical cord mesenchymal stromal cells to target pancreatic cancer cells carrying KRAS mutation. Experiments are run with appropriate controls and data presented is extensive with corroborative evidence.

-Minor formatting issue such as underlined paragraph on page 2, 4, 6 and throughout the manuscript. Seems like review tracking from word document carried over to submission.

Author Response

Dear reviewer,

We greatly appreciate the time you have taken to review our manuscript.

Thank you for this comment. As requested, we made these changes.

Reviewer 3 Report

The authors of the above-referenced manuscript have adequately answered the referees' questions and carefully revised the manuscript. The current revision is a better, improved and readable version, and it can be published without any changes.

Author Response

Thank you for this comment.

Round 3

Reviewer 1 Report

The authors addressed my concerns adequately.